# Residual Platelet Reactivity and Dyslipidemia in Post-CABG Patients Undergoing Repeat Revascularization: Insights from Kazakhstan

**DOI:** 10.3390/diseases13110365

**Published:** 2025-11-09

**Authors:** Aisulu Mussagaliyeva, Sholpan Zhangelova, Laura Danyarova, Friba Nurmukhammad, Dina Kapsultanova, Orazbek Sakhov, Farida Rustamova, Akhmetzhan Sugraliyev, Dana Akhmentayeva

**Affiliations:** 1Department of Postgraduate and Continuing Education, Scientific Research Institute of Cardiology and Internal Diseases, Almaty 050000, Kazakhstan; aisulu.musagalieva@gmail.com (A.M.); zhangelova.s@kaznmu.kz (S.Z.); lbdanyarova@mail.ru (L.D.); ahmentaeva.d@kaznmu.kz (D.A.); 2Faculty of Postgraduate Education, Asfendiyarov Kazakh National Medical University, Almaty 050012, Kazakhstan; sakhov.o@kaznmu.kz (O.S.); rustamova.f@kaznmu.kz (F.R.); sugraliyev.a@kaznmu.kz (A.S.); 3Faculty of Postgraduate Medical Education, Hodja Ahmed Yasawi International Kazakh-Turkish University, Turkestan 161200, Kazakhstan

**Keywords:** coronary artery bypass grafting, residual platelet reactivity, dyslipidemia, repeat revascularization, dual antiplatelet therapy, Kazakhstan

## Abstract

Background: Coronary artery bypass grafting (CABG) remains a standard revascularization strategy for patients with advanced coronary artery disease (CAD). However, a considerable proportion of patients experience recurrent ischemia requiring repeat revascularization. Residual platelet reactivity (RPR) and dyslipidemia are recognized as key factors contributing to graft failure and disease progression. Methods: This observational study was conducted at a tertiary cardiology center in Kazakhstan. A total of 195 post-CABG patients who underwent repeat coronary angiography between 2023 and 2024 recruitment period for recurrent ischemic symptoms within 6–36 months after surgery were included. Clinical characteristics, comorbidities, lipid profiles, and antiplatelet response were analyzed. RPR was measured using the VerifyNow P2Y12 assay when available. Dyslipidemia was defined according to the 2019 and 2021 European guidelines. Results: Elevated RPR was identified in 45% of patients (*n* = 90) despite dual antiplatelet therapy (*p* < 0.01). Poor lipid control was frequent among those who underwent repeat percutaneous coronary intervention (PCI), particularly elevated levels of low-density lipoprotein cholesterol (LDL-C) and total cholesterol (*p* < 0.05). Both elevated RPR and dyslipidemia were independently associated with native coronary disease progression and graft failure (RPR: OR = 2.8; 95% CI 1.4–5.6; *p* = 0.003; dyslipidemia: OR = 2.2; 95% CI 1.1–4.3; *p* = 0.02). The use of ezetimibe was independently associated with a significantly lower risk of repeat stenting (OR = 0.12; 95% CI 0.02–0.75; *p* = 0.023). Smokers were younger, had lower blood pressure, and less frequently presented with diabetes or chronic kidney disease, demonstrating a pattern consistent with the “smoker’s paradox.” Conclusions: Residual platelet reactivity and dyslipidemia are common and clinically relevant predictors of repeat revascularization after CABG. Optimization of antiplatelet and lipid-lowering therapy should be prioritized in secondary prevention for this high-risk population. These findings are particularly important in Kazakhstan, where post-CABG management strategies warrant further improvement.

## 1. Introduction

Coronary artery bypass grafting (CABG) remains a cornerstone in the management of advanced coronary artery disease (CAD), especially in patients with multivessel involvement. While surgical revascularization provides symptomatic relief and mortality benefit, a considerable proportion of patients continue to experience recurrent ischemic symptoms and adverse cardiovascular events during follow-up.

Platelets play a central role in the pathogenesis of atherosclerosis and arterial thrombosis. High residual platelet reactivity (HRPR), even under antiplatelet therapy, is a well-established predictor of major adverse cardiovascular events, including myocardial infarction (MI), stroke, and stent thrombosis [1,2]. Patients with multivessel CAD undergoing CABG have been shown to exhibit higher levels of HRPR compared to those with single-vessel disease. Several comorbidities—including dyslipidemia, anemia, chronic kidney disease (CKD), reduced left ventricular ejection fraction (LVEF), and overweight/obesity—are strongly associated both with HRPR and with the presence of extensive coronary disease, thereby amplifying the risk of recurrent ischemic events [3,4].

The prothrombotic role of platelets is particularly important in patients receiving long-term antiplatelet therapy aimed at reducing MI and stroke incidence. Anemia, common among patients with chronic heart failure (CHF), is linked to impaired response to clopidogrel, increased HRPR, higher hospitalization rates, and worse survival [5,6,7]. Renal dysfunction further diminishes antiplatelet efficacy and contributes to poor outcomes [8]. HRPR has consistently been recognized as a poor prognostic marker in acute coronary syndrome (ACS) [9]. Smoking, dyslipidemia, and elevated cholesterol levels additionally enhance platelet activation and thrombotic risk [4,5]. Moreover, obesity—particularly class I—has been associated with increased HRPR during antiplatelet therapy [10].

In the post-CABG setting, hemostatic alterations induced by cardiopulmonary bypass may further influence platelet function and predispose to enhanced platelet activation [11]. The pathophysiological mechanism linking HRPR to cardiovascular complications primarily involves exaggerated platelet aggregation and thrombus formation [12,13,14]. Consequently, HRPR is considered an independent risk factor for recurrent ischemic events, including MI, graft failure, stent thrombosis, and cardiovascular death [15,16].

Importantly, dyslipidemia represents a modifiable determinant of both platelet hyperreactivity and atherosclerotic progression. Failure to achieve guideline-recommended low-density lipoprotein cholesterol (LDL-C) targets—particularly <1.4 mmol/L in very-high-risk patients according to the 2019/2021 ESC guidelines—significantly increases the risk of recurrent ischemia and need for repeat revascularization. Thus, strict lipid control remains a cornerstone of secondary prevention in post-CABG patients and is crucial for improving long-term outcomes.

The primary objective of our study was to investigate the real clinical and epidemiological profile of patients with a history of coronary artery bypass grafting (CABG) who developed recurrent angina, and to identify significant risk factors associated with angina recurrence or repeat revascularization in this population. Specifically, we analyzed these patients to identify the predominant clinical phenotypes associated with these outcomes, with particular emphasis on dyslipidemia and inadequate LDL-C control, residual platelet reactivity despite dual antiplatelet therapy, anemia, CKD, and heart failure (HF).

## 2. Materials and Methods

### 2.1. Study Design and Setting

This observational and non-experimental cross-sectional non-randomized single center study was conducted from January 2023 to December 2024 recruitment period for recurrent ischemic symptoms within 6–36 months after CABG were included at the Research Institute of Cardiology and Internal Medicine, a tertiary-level cardiovascular center in Almaty, Kazakhstan, which routinely receives complex cases of post-CABG patients from across the country, often with recurrent ischemia resistant to medical therapy. Consequently, all patients included in the study had documented recurrent ischemia after CABG. Patients without angina after CABG were not represented in our hospital population. Our study involved a single assessment of patients as they were admitted over the course of one year, without follow-up. Within this population of patients with recurrent ischemia after CABG, we analyzed clinical and epidemiological characteristics, lipid profiles, and residual platelet reactivity. Depending on the presence of atherosclerosis progression and the level of residual platelet reactivity, patients were divided into two groups. The comparison between these groups allowed us to explore the association between elevated residual platelet reactivity and clinical-metabolic features.

### 2.2. Patient Population

In this study, we analyzed 200 post-CABG patients from a tertiary cardiac center in Kazakhstan. Recurrent angina was observed in 94% (*n* = 188), and 68% (*n* = 136) required repeat percutaneous coronary intervention (PCI) after coronary angiography. Eligible participants were adult patients (≥18 years) with a documented history of coronary artery bypass grafting (CABG) and a diagnosis of stable coronary artery disease (CAD) who were admitted for repeat coronary angiography due to recurrent ischemic symptoms.

### 2.3. Inclusion Criteria

Prior history of CABGDocumented stable CADClinical and/or instrumental evidence of recurrent myocardial ischemia (e.g., angina, positive stress test, ischemic ECG changes)

### 2.4. Exclusion Criteria

Presence of atrial fibrillationCurrent use of oral anticoagulant therapy (e.g., rivaroxaban, apixaban, dabigatran)Diagnosis of acute coronary syndrome (ACS) at the time of admissionA detailed flowchart of patient selection, including inclusion and exclusion criteria, is presented in Figure 1.

### 2.5. Residual Platelet Reactivity Assessment

Residual platelet reactivity (RPR) was evaluated using the VerifyNow P2Y12 assay (Instrumentation Laboratory, Bedford, MA, USA), a point-of-care system that quantifies platelet inhibition and reports results as P2Y12 Reaction Units (PRU). High residual platelet reactivity (HRPR) was defined as PRU ≥ 208, in accordance with current international consensus guidelines.

### 2.6. Clinical and Laboratory Data Collection

Baseline demographic, clinical, and pharmacological characteristics were extracted from electronic medical records. The following parameters were systematically evaluated:Lipid profile: Total cholesterol, LDL-C, HDL-C, triglycerides.Renal function: Serum creatinine and estimated glomerular filtration rate (eGFR) to identify chronic kidney disease (CKD).Hemoglobin level: Assessed for anemia according to WHO criteria.Left ventricular ejection fraction (LVEF): Measured via transthoracic echocardiography.Body mass index (BMI): Used to classify overweight and obesity.Cardiovascular complications: Including heart failure, diabetes mellitus, and prior myocardial infarction.

All patients received high-intensity doses of atorvastatin 40 mg or rosuvastatin 20 mg, and only one-third of the patients were taking ezetimibe 10 mg (which is not pro-vided free of charge by prescription in Kazakhstan) to achieve an LDL-C level below 1.4 mmol/L, as well as antiplatelet therapy, in accordance with the current 2024 European Society of Cardiology (ESC) guidelines for the management of chronic coronary syndromes [17].

### 2.7. Ethical Considerations

The study was conducted in accordance with the ethical principles of the Declaration of Helsinki. Ethical approval was obtained from the Local Ethics Committee of Asfendiyarov Kazakh National Medical University, Almaty, Republic of Kazakhstan (Protocol No. 11 (134); Date: 4 November 2022). Written informed consent was obtained from all participants prior to enrollment.

### 2.8. Statistical Analysis

The data are presented as Median [Quartile 1; Quartile 3], as the majority of the quantitative variables do not follow a normal distribution. Between-group comparisons were performed using the Mann–Whitney U test for quantitative variables, or the chi-squared test (Fisher’s exact test when the number of observations was small) for qualitative variables. Categorical variables were expressed as frequencies and percentages. Normality of continuous variables, including PRU values, was assessed using the Shapiro–Wilk test. For comparisons of continuous variables between groups (e.g., HRPR vs. non-HRPR) Mann–Whitney U test was used as appropriate. Categorical variables were analyzed using the Chi-square test or Fisher’s exact test. A stepwise logistic regression analysis was performed to identify predictors associated with the risk of stenting after coronary artery bypass grafting (CABG). A *p*-value < 0.05 was considered statistically significant. All statistical tests were two-sided, and 95% confidence intervals (CIs) were reported when relevant.

## 3. Results

In this study, we analyzed 195 post-CABG patients from a tertiary cardiac center in Kazakhstan. Recurrent angina was observed in 94% (*n* = 188), and 68% (*n* = 136) required repeat percutaneous coronary intervention (PCI) after coronary angiography.

The mean age of the study population was 63.4 ± 7.56 years. Regarding sex distribution, men predominated, accounting for 73% (*n* = 146) of the group, while women represented 27% (*n* = 54).

Diabetes mellitus was present in 31.5% of patients (*n* = 63), prior myocardial infarction in 53% (*n* = 106), and previous stroke in 8% (*n* = 16). All-cause mortality during follow-up occurred in 2.5% (*n* = 5). Left ventricular aneurysm was identified in 14% (*n* = 28) of patients, and a history of coronary stenting was reported in 17% (*n* = 34). Among behavioral risk factors, smoking was documented in 36.5% (*n* = 73) and alcohol consumption in 2.5% (*n* = 5). The majority of participants were overweight or obese, with a body mass index (BMI) > 25 kg/m^2^ observed in 74.5% (*n* = 149). Chronic kidney disease (CKD) was diagnosed in 44.5% (*n* = 89) of patients. The mean estimated glomerular filtration rate (eGFR) was 82 ± 16.4 mL/min/1.73 m^2^. Based on KDIGO classification, CKD stage I (eGFR > 90 mL/min/1.73 m^2^) was identified in 3.4% (*n* = 3), stage II (eGFR 60–89) in 76% (*n* = 68), stage III (eGFR 30–59) in 19% (*n* = 17), and stage IV (eGFR 15–29) in 1.6% (*n* = 1). No cases of stage V CKD (eGFR < 15 mL/min/1.73 m^2^) were reported.

Among 195 patients, 136 (69.7%) underwent repeat percutaneous coronary intervention (PCI), whereas 59 (30.3%) did not require reintervention due to the following reasons: absence of hemodynamically significant stenoses, technically unfeasible stenting due to chronic occlusions or distal segment lesions, high procedural risk, microvascular angina and comorbidities. These patients were managed with optimal medical therapy under close follow-up. Baseline demographic, clinical, laboratory, and instrumental characteristics were systematically compared between the two groups (Table 1). A comparative analysis was performed within the group of patients with recurrent angina after CABG, who were divided into two subgroups according to the presence or absence of re-stenting.

The full scope of the research and statistical data is presented in Appendix A.

### 3.1. Demographic and Clinical Characteristics in Groups with and Without Restenting

Patients requiring repeat PCI were significantly older than those without reintervention (median age: 71.0 [IQR 65.8–75.0] vs. 66.0 [61.5–73.0] years; *p* = 0.016). No significant difference in sex distribution was observed (*p* = 0.431). Current smoking was more frequent in the non-reintervention group (54.2% vs. 28.7%; *p* = 0.001), whereas alcohol consumption rates were low and comparable (*p* = 0.164). Systolic blood pressure was higher in the repeat PCI group (130 [120–140] vs. 120 [110–130] mmHg; *p* = 0.041), while diastolic pressure did not differ significantly (Table 1).

### 3.2. Comorbidities and Medical History

The prevalence of hypertension, diabetes mellitus, prior myocardial infarction, stroke, chronic kidney disease (CKD), anemia, and obesity was similar between groups. However, a higher frequency of prior CABG was noted among patients undergoing reintervention (96.3% vs. 83.1%; *p* = 0.003).

### 3.3. Biochemical and Hematological Parameters

No significant group differences were found in lipid profile (total cholesterol, LDL-C, HDL-C, triglycerides), glycemic control (glucose, HbA1c), renal function (creatinine, eGFR), liver enzymes (ALT, AST), inflammatory markers (CRP), or thyroid function (TSH, T4). Interestingly, cardiac troponin levels were lower in the reintervention group (14.3 [10.3–32.8] vs. 20.0 [13.4–95.4] ng/L; *p* = 0.002), possibly reflecting differences in post-CABG myocardial injury dynamics.

### 3.4. Platelet Reactivity

A significantly higher platelet reactivity was observed in patients who underwent re-stenting compared to those without re-stenting. The median PRU was 230 [230–274] versus 145 [140–155] (*p* < 0.001). High residual platelet reactivity (HRPR) was detected in 66.2% of patients in the re-stenting group, while no cases were observed in the non–re-stenting group (*p* < 0.001).

### 3.5. Echocardiographic Findings

End-diastolic volume (EDV) was lower in the reintervention group (136 [113–179] vs. 162 [129–206] mL; *p* = 0.038). Posterior wall thickness (PWT) was also higher (1.10 [1.00–1.20] vs. 1.05 [0.90–1.20] cm; *p* = 0.030). No significant differences were observed for LVEF, stroke volume, left atrial/ventricular dimensions, or pulmonary artery pressure (Table 2, Appendix A).

### 3.6. Coronary Angiographic Findings

Patients with repeat PCI demonstrated more extensive coronary artery disease. Significant stenosis was more frequently detected in:LAD: 52.9% vs. 27.1% (*p* = 0.002)LCx: 37.5% vs. 13.6% (*p* = 0.002)Diagonal branch: 30.9% vs. 8.5% (*p* = 0.001)PDA: 16.9% vs. 1.7% (*p* = 0.006)SVGs: 25.0% vs. 5.1% (*p* = 0.002)

The number of bypass grafts (3.00 [2.00–3.00] vs. 2.00 [2.00–3.00]; *p* < 0.001) and diseased vessels (4.00 [3.75–5.00] vs. 3.00 [2.00–3.00]; *p* < 0.001) was also higher in the reintervention group.

### 3.7. Medical Therapy in Groups with and Without Restenting

All patients received statin therapy (atorvastatin 40 mg, rosuvastatin 20 mg). Ezetimibe 10 mg was prescribed more frequently in the reintervention group (34.6% vs. 0%; *p* < 0.001), suggesting intensified lipid-lowering strategies in patients with progressive disease.

### 3.8. Symptoms and Outcomes

Among patients with recurrent angina, clinical manifestations were observed in 88.1% (*p* < 0.001) of those without re-stenting and in 100% of those with re-stenting. In seven patients from the non–re-stenting group, dyspnea was identified as the main manifestation of myocardial ischemia and was considered an anginal equivalent.

### 3.9. Summary of Key Findings

High residual platelet reactivity was strongly associated with repeat revascularization during one year follow-up period.

Older age, greater coronary involvement, and higher number of bypass grafts predicted the need for reintervention.

Despite similar baseline lipid levels, patients requiring repeat PCI were more frequently prescribed ezetimibe.

The prevalence of anemia, CKD, and diabetes was similar between groups, suggesting HRPR and coronary disease burden play a more central role in post-CABG ischemic recurrence.

The comparison was performed within the group of patients with recurrent angina after CABG, divided into two subgroups according to the level of residual platelet reactivity (normal vs. high).

Based on this stratification, we analyzed their clinical and epidemiological characteristics and angiographic findings. Thus, our study design can be more accurately described as a comparative, observational, non-randomized study of consecutively included patients with recurrent angina after CABG.

A total of 195 patients with a history of coronary artery bypass grafting (CABG) were included in the analysis. Based on platelet reactivity (PRU) and clinical criteria, patients were stratified into two groups: those who achieved the therapeutic window for antiplatelet therapy (*n* = 105) and those classified as having a high residual thrombotic risk (HRTR; *n* = 90). Comparative analyses of demographic, clinical, laboratory, and angiographic characteristics were performed (Table 3). The full scope of the research and statistical data is presented in Appendix A.

### 3.10. Demographic and Clinical Characteristics Depending on Residual Platelet Reactivity

No significant differences were observed between the groups regarding age (69.0 [62.0–74.0] vs. 70.5 [65.2–75.0] years; *p* = 0.147), sex distribution (*p* = 0.871), or body mass index (BMI; *p* = 0.096). Interestingly, the prevalence of current smoking was lower in the HRTR group compared with the therapeutic window group (27.8% vs. 43.8%; *p* = 0.030). Patients in the HRTR group exhibited significantly higher systolic blood pressure (130 [120–140] vs. 120 [115–130] mmHg; *p* = 0.031) and fasting glucose levels (5.90 [5.23–7.10] vs. 5.54 [5.00–6.20] mmol/L; *p* = 0.022). A prior history of stroke was also more frequent in the HRTR group (13.3% vs. 3.81%; *p* = 0.031).

### 3.11. Angiographic Findings

Patients with HRTR had more extensive coronary artery disease, with significantly higher rates of multivessel involvement. Severe stenoses were more prevalent in the proximal left anterior descending artery (57.8% vs. 34.3%; *p* = 0.002), left circumflex artery (42.2% vs. 20.0%; *p* = 0.001), diagonal branches (34.4% vs. 15.2%; *p* = 0.003), posterior descending artery (21.1% vs. 4.76%; *p* = 0.001), and venous bypass grafts (33.3% vs. 6.67%; *p* < 0.001). Furthermore, the HRTR group had a higher number of diseased vessels (4.00 [4.00–5.00] vs. 3.00 [2.00–3.00]; *p* < 0.001) and bypass grafts (3.00 [2.00–4.00] vs. 2.00 [2.00–3.00]; *p* = 0.001).

### 3.12. Platelet Reactivity and Clinical Outcomes

Platelet reactivity was markedly elevated in the HRTR group (PRU: 255 [230–280] vs. 170 [141–200]; *p* < 0.001), indicating a suboptimal response to dual antiplatelet therapy. Consistently, all patients in the HRTR group underwent post-CABG stenting (100% vs. 43.8%; *p* < 0.001) and experienced recurrent angina (100% vs. 93.3%; *p* = 0.016).

### 3.13. Medical Therapy in Groups with HRPR 

All patients were treated with statins (atorvastatin 40 mg or rosuvastatin 20 mg); however, ezetimibe was prescribed significantly more often in the HRTR group (47.8% vs. 3.81%; *p* < 0.001), reflecting attempts at intensified lipid-lowering therapy. Patients after repeat revascularization were administered dual antiplatelet therapy (acetylsalicylic acid and clopidogrel).

### 3.14. Summary of Findings


Patients classified as HRTR exhibited:More extensive and severe coronary atherosclerosis;Higher platelet reactivity despite dual antiplatelet therapy;Increased frequency of recurrent angina and need for repeat revascularization;Greater use of adjunctive lipid-lowering therapy.


These findings emphasize the role of HRTR as a marker of poor outcomes in post-CABG patients and underscore the importance of individualized antiplatelet and lipid-lowering strategies in this high-risk population.

A stepwise logistic regression analysis was performed to identify predictors associated with the risk of stenting after coronary artery bypass grafting (CABG). Due to the relatively small sample size (*n* = 195) and a large number of potential covariates, a stepwise forward/backward selection algorithm was employed (Table 4).

Among the variables included in the regression analysis, the number of affected vessels emerged as the strongest and statistically significant predictor of post-CABG stenting. Patients with more extensive coronary involvement had markedly higher odds of requiring stent implantation (Odds Ratio [OR] = 52.67; 95% CI: 17.33–213.77; *p* < 0.001).

Conversely, the use of ezetimibe was independently associated with a significantly lower risk of stenting (OR = 0.12; 95% CI: 0.02–0.75; *p* = 0.023), suggesting a potential protective effect of this lipid-lowering therapy in the postoperative setting.

Other factors, including thyroid-stimulating hormone (TSH) levels (OR = 0.87; 95% CI: 0.72–1.00; *p* = 0.116), serum potassium (OR = 2.53; 95% CI: 0.78–8.59; *p* = 0.127), and glycated hemoglobin (HbA1c) (OR = 1.18; 95% CI: 0.95–1.46; *p* = 0.138), did not reach statistical significance but were retained in the final model as clinically relevant covariates.

The regression model demonstrated good explanatory power, with a Tjur’s R^2^ of 0.681, indicating that a substantial proportion of the variance in stenting risk was explained by the selected predictors.

Collectively, these findings emphasize the critical role of disease burden in determining the need for repeat revascularization, while also highlighting the potential benefit of ezetimibe as part of intensified secondary prevention strategies in post-CABG patients.

## 4. Discussion

Our study highlights the substantial burden of recurrent ischemia and the high rate of re-intervention among post-CABG patients in a real-world clinical setting. In patients after CABG, recurrent angina symptoms developed within 6–36 months. Repeat coronary angiography (CAG) was performed in 94% (*n* = 188) of patients, of whom 68% underwent re-stenting. It should be noted that these data do not reflect the population incidence of recurrent angina, as patients were referred from various regions to a tertiary center for the study. However, our aim was to characterize the phenotypic features and underlying causes of angina in this high-risk group of patients with recurrent angina following CABG. These findings reflect the ongoing progression of atherosclerosis and/or incomplete revascularization, underscoring the challenges of long-term secondary prevention in this population. The role of platelets in arterial thrombosis is well established, particularly in patients with coronary heart disease (CHD) who are on antiplatelet therapy, which has been shown to reduce the risk of major adverse cardiovascular events. Hemoglobin levels also serve as a critical prognostic marker in patients with chronic heart failure (CHF), where anemia is a frequent comorbidity and is associated with increased risks of hospitalization and mortality [2]. Patients with HRPR who undergo percutaneous coronary intervention (PCI) with stent placement while receiving clopidogrel therapy are at elevated risk for recurrent ischemic events. Notably, residual platelet reactivity and HRPR levels have been found to increase in parallel with declining kidney function [3]. Smoking, a well-known risk factor for CHD and MI, is also strongly associated with increased platelet reactivity. When combined with HRPR, smoking further elevates the risk of stent thrombosis [4]. The our analysis of smokers and non-smokers showed that smokers were younger, had lower blood pressure, less frequently had diabetes mellitus and chronic kidney disease, underwent CABG less often, had fewer vessels involved in the pathological process, and showed a lower degree of vascular damage. Thus, it is likely that we are observing the “smoker’s paradox.” It can be suggested that there are conditions for less frequent use of stenting among patients in this subgroup. In addition, there is an assumption that patients who underwent stenting were more likely to quit smoking. Dyslipidemia contributes to heightened platelet reactivity and thrombotic risk. Elevated cholesterol levels are known to enhance platelet activation and aggregation [5]. Similarly, the extent of coronary atherosclerosis correlates positively with platelet reactivity, suggesting that increased HRPR reflects more widespread atherosclerotic involvement [6]. Anemia has been shown to impair responsiveness to clopidogrel, thus contributing to elevated HRPR [7], while renal dysfunction is also independently associated with increased platelet reactivity [8]. HRPR is recognized as a poor prognostic marker in acute coronary syndrome (ACS) [9]. Moreover, grade 1 obesity has been linked to higher HRPR in patients undergoing antiplatelet therapy [10]. In patients undergoing CABG, alterations in the hemostatic system due to cardiopulmonary bypass may further contribute to increased platelet reactivity [11]. The primary association between HRPR and cardiovascular risk is mediated through enhanced platelet activation, which accelerates thrombus formation [12]. HRPR is therefore regarded as an independent risk factor for a range of serious complications, including myocardial infarction, stent thrombosis, and cardiovascular death [13]. Platelets are directly involved in thrombus formation following the rupture of atherosclerotic plaques in the coronary arteries [14], and individuals with HRPR are at significantly higher risk of recurrent ischemic events [15].

Despite adherence to the 2024 European Society of Cardiology (ESC) guidelines for the management of chronic coronary syndromes, a considerable proportion of patients continued to experience ischemic symptoms. This suggests the persistence of residual risk factors, including dyslipidemia and high platelet reactivity, as well as difficulties in achieving recommended LDL-C targets.

A key driver of recurrent ischemia was persistent dyslipidemia, with most patients failing to reach LDL-C goals despite statin therapy. Limited access to more potent lipid-lowering therapies (e.g., ezetimibe, PCSK9 inhibitors) further contributed to this gap. In Kazakhstan, these medications are not included in the national reimbursement list and are therefore not provided free of charge, which likely explains the underutilization of combination lipid-lowering therapy and the near-universal failure to achieve LDL-C targets [16].

In addition to biochemical and platelet-derived determinants of residual risk, Sonaglioni et al. [18] reported that thoracic conformation itself may influence cardiovascular prognosis, as patients with a concave-shaped chest wall exhibited a significantly lower incidence of cardiovascular events.

The present study demonstrated a notably higher rate of angina recurrence compared to data reported in the international literature. In our cohort, recurrent angina was observed in 45% of patients, whereas most contemporary studies report rates ranging from 10% to 25% within similar follow-up periods after myocardial revascularization (both PCI and CABG) [19,20,21]. This discrepancy may be explained by several factors, including population characteristics such as a high prevalence of residual platelet reactivity and suboptimal lipid control despite intensive therapy. Moreover, the rate of graft failure after CABG was relatively high, which likely contributed to symptom recurrence. Limited availability of certain lipid-lowering agents (for instance, ezetimibe is not routinely prescribed in Kazakhstan) and potential differences in adherence to secondary prevention strategies might also play a role. These findings highlight the need for more aggressive risk factor modification, optimized lipid and antiplatelet therapy, and closer long-term follow-up in this patient population. This observation underscores the importance of comprehensive secondary prevention programs and individualized therapy optimization to reduce graft failure and recurrent ischemic events in post-revascularization patients [22].

Another important finding was the high prevalence of residual platelet reactivity, as measured by aggregometry in patients receiving dual antiplatelet therapy. This phenotype is known to be associated with increased thrombotic risk, recurrent ischemia, and a higher likelihood of stent failure, which is consistent with our observation of frequent post-CABG revascularization. In addition, comorbid conditions such as anemia, chronic kidney disease, and heart failure were common and likely amplified residual cardiovascular risk. These results are in line with previous reports demonstrating that non-cardiac comorbidities significantly impair long-term outcomes after surgical revascularization. Taken together, our findings point to distinct clinical phenotypes—such as dyslipidemic-thrombotic or cardio-renal-anemic—that may benefit from individualized post-operative management rather than uniform treatment protocols. This approach is particularly relevant in underrepresented regions such as Central Asia, where healthcare resource limitations further complicate secondary prevention.

## 5. Limitations

This study has several limitations. First, it was a retrospective, single-center analysis conducted at the National Research Institute of Cardiology and Internal Medicine, Al-maty, Kazakhstan. The study was conducted at a tertiary-care referral center, which routinely receives complex cases of post-CABG patients from across the country, often with recurrent ischemia resistant to medical therapy. Consequently, all patients included in the study had documented recurrent ischemia after CABG. The high rate of angina recurrence observed herein reflects the specific characteristics of the tertiary-care referral population, rather than the prevalence of this condition in the general post-CABG population.

However, only patients with signs and symptoms of ischemia were included.

One of the main limitations of this study is the relatively small sample size, which made it impossible to construct a robust multivariate model. The model developed using stepwise predictor selection produced imbalanced results, limiting its applicability for predicting specific outcomes. Nevertheless, it can provide valuable insights into the presence and direction of associations between independent and dependent variables.

Additionally, key parameters such as inflammatory biomarkers or genetic determi-nants of platelet resistance were not consistently available due to real-world practice constraints.

## 6. Conclusions

Residual dyslipidemia and platelet hyperreactivity remain prevalent and clinically significant contributors to recurrent ischemia and repeat revascularization after CABG. Patients requiring repeat PCI had more extensive coronary involvement, elevated platelet reactivity reflecting suboptimal antiplatelet response, and were more frequently prescribed intensified lipid-lowering therapy. Importantly, the extent of vascular disease and ezetimibe use emerged as independent predictors of stenting risk. These results underscore the need for personalized strategies in lipid-lowering and antiplatelet therapy to optimize outcomes in this high-risk population. Tailored phenotype-guided approaches are especially critical in Kazakhstan, where limited access to advanced therapies may exacerbate residual risk and impede secondary prevention efforts.

## Figures and Tables

**Figure 1 diseases-13-00365-f001:**
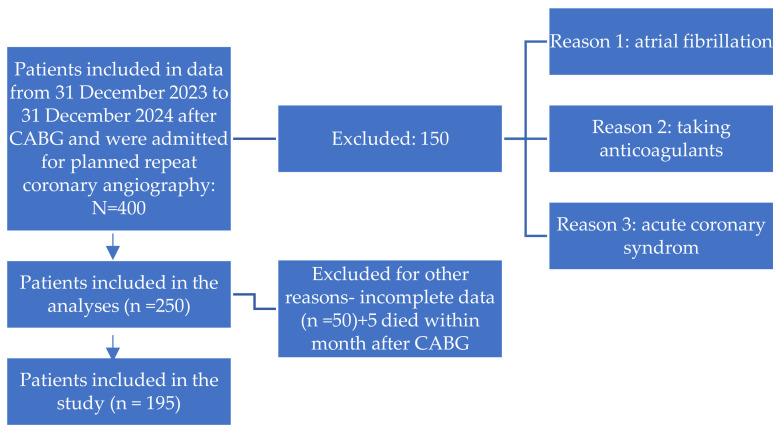
The flow chart of patients.

**Table 1 diseases-13-00365-t001:** Clinical and anamnestic characteristics of the patients.

Parameter	Without Re-Stenting *n* = 59	With Re-Stenting *n* = 136	*p*-Value
Age	66.0 [61.5; 73.0]	71.0 [65.8; 75.0]	0.016
Gender:			0.431
female	13 (22.0%)	39 (28.7%)	
male	46 (78.0%)	97 (71.3%)	
Smoking:			0.001
No	27 (45.8%)	97 (71.3%)	
Yes	32 (54.2%)	39 (28.7%)	
Alcohol:			0.164
No	56 (94.9%)	134 (98.5%)	
Yes	3 (5.08%)	2 (1.47%)	
Weight	79.0 [70.0; 86.5]	78.5 [70.0; 85.8]	0.903
Height	170 [164; 174]	168 [162; 173]	0.438
BMI	27.3 [24.9; 29.4]	27.6 [25.0; 30.2]	0.411
SBP	120 [110; 130]	130 [120; 140]	0.041
DBP	80.0 [70.0; 80.0]	80.0 [70.0; 80.0]	0.201
History of myocardial infarction:			0.145
No	33 (55.9%)	59 (43.4%)	
Yes	26 (44.1%)	77 (56.6%)	
Arterial hypertension:			0.133
No	5 (8.47%)	4 (2.94%)	
Yes	54 (91.5%)	132 (97.1%)	
DM type 2:			0.603
No	42 (71.2%)	90 (66.2%)	
Yes	17 (28.8%)	46 (33.8%)	
History of Stroke:			0.400
No	56 (94.9%)	123 (90.4%)	
Yes	3 (5.08%)	13 (9.56%)	
CKD:			0.068
No	39 (66.1%)	69 (50.7%)	
Yes	20 (33.9%)	67 (49.3%)	
Stenting:			0.712
No	47 (79.7%)	113 (83.1%)	
Yes	12 (20.3%)	23 (16.9%)	
IHD, duration	3.00 [1.00; 6.50]	2.50 [1.00; 10.0]	0.364
CABG:			0.003
No stents before CARB	10 (16.9%)	5 (3.68%)	
Yes—there were stents before CABG	49 (83.1%)	131 (96.3%)	

**Table 2 diseases-13-00365-t002:** Descriptive statistics of the group.

Parameter	Without Re-Stenting *n* = 59	With Re-Stenting *n* = 136	*p*-Value
Hemoglobin	142 [132; 154]	145 [135; 155]	0.542
CRP	2.00 [1.24; 3.56]	2.45 [1.03; 4.22]	0.487
Glucose	5.54 [5.12; 6.30]	5.80 [5.16; 6.93]	0.293
HbA1c	6.01 [5.30; 6.70]	6.05 [5.40; 7.33]	0.403
Creatinine	80.4 [70.6; 90.2]	79.0 [70.8; 92.2]	0.952
GFR	88.0 [75.8; 96.0]	81.0 [69.8; 94.2]	0.279
Potassium	4.40 [4.20; 4.60]	4.30 [4.10; 4.60]	0.590
Sodium	142 [140; 144]	142 [141; 144]	0.268
ALT	19.0 [14.1; 28.5]	18.1 [12.9; 25.7]	0.453
AST	17.3 [13.8; 25.9]	17.9 [14.9; 23.1]	0.977
TC	4.24 [3.41; 5.25]	4.50 [3.70; 5.22]	0.419
LDL	2.58 [2.00; 3.17]	2.42 [2.00; 3.20]	0.569
HDL	1.00 [0.99; 1.00]	1.00 [0.96; 1.03]	0.809
TG	1.50 [1.00; 1.98]	1.29 [1.00; 1.75]	0.083
Troponin I	20.0 [13.4; 95.4]	14.3 [10.3; 32.8]	0.002
EF	47.0 [41.5; 59.0]	52.0 [44.0; 59.2]	0.235
LA	3.80 [3.60; 4.55]	3.90 [3.50; 4.40]	0.638
EDD	5.70 [5.20; 6.30]	5.30 [4.97; 6.00]	0.081
ESD	4.20 [3.35; 4.90]	3.80 [3.30; 4.60]	0.240
LVPW	1.05 [0.90; 1.20]	1.10 [1.00; 1.20]	0.030
IVS	1.10 [0.95; 1.27]	1.10 [1.00; 1.30]	0.924
LV aneurysm:			0.990
No	50 (84.7%)	117 (86.0%)	
Yes	9 (15.3%)	19 (14.0%)	
Heart Rate	75.0 [66.0; 82.0]	71.5 [65.0; 83.2]	0.496
Number of shunts	2.00 [2.00; 3.00]	3.00 [2.00; 3.00]	<0.001
Number of affected vessels	3.00 [2.00; 3.00]	4.00 [3.75; 5.00]	<0.001
Statins (high-intensity statins atorvastatin 40 mg or rosuvastatin 20 mg): Yes	59 (100%)	136 (100%)	
Ezetimibe:			<0.001
No	59 (100%)	89 (65.4%)	
Yes	0 (0.00%)	47 (34.6%)	
Angina after CABG:			<0.001
No	7 (11.9%)	0 (0.00%)	
Yes	52 (88.1%)	136 (100%)	
Mortality after CABG:			1.000
No	58 (98.3%)	132 (97.1%)	
Yes	1 (1.69%)	4 (2.94%)	
Stenting after CABG:			<0.001
No	59 (100%)	0 (0.00%)	
Yes	0 (0.00%)	136 (100%)	
PRU	145 [140; 155]	230 [200; 274]	<0.001
Dyslipidemia:			1.000
No	2 (3.39%)	4 (2.94%)	
Yes	57 (96.6%)	132 (97.1%)	
Anemia:			0.941
No	52 (88.1%)	122 (89.7%)	
Yes	7 (11.9%)	14 (10.3%)	
CKD:			0.977
No	53 (89.8%)	124 (91.2%)	
Yes	6 (10.2%)	12 (8.82%)	
Obesity:			0.406
No	17 (28.8%)	30 (22.1%)	
Yes	42 (71.2%)	106 (77.9%)	

**Table 3 diseases-13-00365-t003:** Descriptive statistics of the Therapeutic window/HRPR group.

	Therapeutic Window *n* = 105	HRPR *n* = 90	*p*-Value
Hemoglobin	142 [132; 154]	150 [136; 156]	0.053
CRP	2.36 [1.27; 4.10]	2.29 [1.02; 4.13]	0.853
Glucose	5.54 [5.00; 6.20]	5.90 [5.23; 7.10]	0.022
HbA1c	6.01 [5.38; 6.70]	6.06 [5.52; 7.83]	0.282
Creatinine	80.0 [71.0; 91.8]	79.0 [70.3; 92.7]	0.890
GFR	87.0 [74.0; 96.0]	81.0 [70.0; 92.5]	0.361
Potassium	4.30 [4.10; 4.50]	4.35 [4.10; 4.60]	0.267
Sodium	142 [141; 144]	142 [140; 144]	0.931
ALT	18.0 [12.0; 27.5]	19.0 [14.0; 26.4]	0.473
AST	17.6 [13.8; 25.8]	17.7 [15.0; 22.7]	0.934
TC	4.30 [3.67; 5.27]	4.50 [3.63; 5.20]	0.571
LDL	2.43 [2.00; 3.10]	2.64 [2.00; 3.39]	0.546
HDL	1.00 [0.98; 1.01]	1.00 [0.96; 1.04]	0.684
TG	1.37 [1.00; 1.80]	1.38 [1.00; 1.79]	0.981
Troponin I	15.0 [11.0; 58.7]	15.0 [11.0; 40.0]	0.635
EF	49.0 [41.0; 60.0]	51.7 [45.0; 58.9]	0.459
LV aneurysm:			0.862
No	89 (84.8%)	78 (86.7%)	
Yes	16 (15.2%)	12 (13.3%)	
Heart Rate	74.0 [66.0; 83.0]	70.5 [65.0; 82.0]	0.483
USD of BCA:			0.196
No	48 (45.7%)	32 (35.6%)	
Yes	57 (54.3%)	58 (64.4%)	
Number of shunts	2.00 [2.00; 3.00]	3.00 [2.00; 4.00]	0.001
Number of affected vessels	3.00 [2.00; 3.00]	4.00 [4.00; 5.00]	<0.001
Statins: Yes	105 (100%)	90 (100%)	
Ezetimibe:			<0.001
No	101 (96.2%)	47 (52.2%)	
Yes	4 (3.81%)	43 (47.8%)	
Angina after CABG:			0.016
No	7 (6.67%)	0 (0.00%)	
Yes	98 (93.3%)	90 (100%)	
Mortality after CABG:			0.183
No	104 (99.0%)	86 (95.6%)	
Yes	1 (0.95%)	4 (4.44%)	
Stenting after CABG:			<0.001
No	59 (56.2%)	0 (0.00%)	
Yes	46 (43.8%)	90 (100%)	
Dyslipidemia:			0.688
No	4 (3.81%)	2 (2.22%)	
Yes	101 (96.2%)	88 (97.8%)	
Anemia:			0.310
No	91 (86.7%)	83 (92.2%)	
Yes	14 (13.3%)	7 (7.78%)	
CKD:			0.689
No	94 (89.5%)	83 (92.2%)	
Yes	11 (10.5%)	7 (7.78%)	
Obesity:			0.462
No	28 (26.7%)	19 (21.1%)	
Yes	77 (73.3%)	71 (78.9%)	

**Table 4 diseases-13-00365-t004:** Stepwise logistic regression: predictors associated with the risk of developing stenting after CABG.

	HRPR		
Predictors	Odds Ratios	CI	*p*
Number of affected vessels	52.67	17.33–213.77	<0.001
TSH	0.87	0.72–1.00	0.116
Ezetimibe: Yes	0.12	0.02–0.75	0.023
Potassium	2.53	0.78–8.59	0.127
Hb A 1 c	1.18	0.95–1.46	0.138
Observations	195		
R^2^ Tjur	0.681		

## Data Availability

The raw data supporting the conclusions of this article will be made available by the authors on request.

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
