# Peer review of "Residual Platelet Reactivity and Dyslipidemia in Post-CABG Patients Undergoing Repeat Revascularization: Insights from Kazakhstan"

_diseases, 2025, doi:10.3390/diseases13110365_

Round 1
Reviewer 1 Report
Comments and Suggestions for Authors
In this interesting study, the authors aimed at investigating the association of residual platelet reactivity (RPR) and dyslipidemia with repeat revascularization in post-CABG patients.
The authors found that poor lipid control, especially increased low-density lipoprotein cholesterol (LDL-C) and total cholesterol was common among patients undergoing repeat percutaneous coronary intervention (PCI). Both RPR and dyslipidemia were independently associated with native coronary disease progression and graft failure.
Accordingly, the authors concluded that optimizing antiplatelet and lipid-lowering therapy should be a priority in secondary prevention for this high-risk group.
The authors findings point to distinct clinical phenotypes—such as dyslipidemic-thrombotic or cardio-renal-anemic—that may benefit from individualized post-operative management rather than uniform treatment protocols.
The manuscript provides valuable insight into the combined prognostic impact of residual platelet reactivity and dyslipidemia in post-CABG patients, addressing a critical gap in regional and international literature. By linking impaired antiplatelet response and lipid control to repeat revascularization risk, this study enhances understanding of secondary prevention strategies in underrepresented Central Asian populations.
The manuscript is clearly and coherently written, demonstrating a logical flow from background to conclusions. The methodological design is rigorous and appropriately detailed, ensuring reproducibility of the findings. Tables and figures are well structured and effectively illustrate the key results. The references are relevant and up to date, and the conclusions are sound, being well supported by the presented data and statistical analyses.
I have only one suggestion for the authors.
In the Discussion section, the authors could also highlight the potential usefulness of innovative prognostic indicators that may assist clinicians in the risk stratification of patients with coronary artery disease (CAD). In particular, anthropometric parameters such as chest wall conformation merit consideration. Recent evidence suggests that individuals with a narrow anteroposterior thoracic diameter and/or varying degrees of anterior chest deformities, including pectus excavatum, exhibit increased symptom perception and more favorable mid- to long-term outcomes. The following references could be emphasized: PMID: 37491452 and PMID: 34485034.
Author Response
Comment 1: In the Discussion section, the authors could also highlight the potential usefulness of innovative prognostic indicators that may assist clinicians in the risk stratification of patients with coronary artery disease (CAD). In particular, anthropometric parameters such as chest wall conformation merit consideration. Recent evidence suggests that individuals with a narrow anteroposterior thoracic diameter and/or varying degrees of anterior chest deformities, including pectus excavatum, exhibit increased symptom perception and more favorable mid- to long-term outcomes. The following references could be emphasized: PMID: 37491452 and PMID: 34485034.
Response:
We are very grateful to you for reviewing our article.
According to your suggestion: In the Discussion section, the authors could also highlight the potential usefulness of innovative prognostic indicators that may assist clinicians in the risk stratification of patients with coronary artery disease (CAD). In particular, anthropometric parameters such as chest wall conformation merit consideration. Recent evidence suggests that individuals with a narrow anteroposterior thoracic diameter and/or varying degrees of anterior chest deformities, including pectus excavatum, exhibit increased symptom perception and more favorable mid- to long-term outcomes. The following references could be emphasized: PMID: 37491452 and PMID: 34485034.
- In the study you cited — PMID: 37491452
Hohneck A, Ansari U, Natale M, et al. Description of a new clinical syndrome: thoracic constriction without evidence of the typical funnel-shaped depression — the "invisible" pectus excavatum. Sci Rep. 2023;13(1):12036. Published 2023 Jul 25. doi:10.1038/s41598-023-38739-w — the investigated patients had a mean age of 35 ± 16 years. This cohort is not comparable to the elderly population represented in our study; therefore, we did not include it in our discussion.
- We use the data from the second article PMID: 34485034: Sonaglioni A, Rigamonti E, Nicolosi GL, Lombardo M. Prognostic Value of Modified Haller Index in Patients with Suspected Coronary Artery Disease Referred for Exercise Stress Echocardiography. J Cardiovasc Echogr. 2021;31(2):85-95. doi:10.4103/jcecho.jcecho_141_20
We added in discussion this text:
In addition to biochemical and platelet-derived determinants of residual risk, Sonaglioni et al. [18] reported that thoracic conformation itself may influence cardiovascular prognosis, as patients with a concave-shaped chest wall exhibited a significantly lower incidence of cardiovascular events.
P_13__, Line__381-384_
And added:
- Sonaglioni A, Rigamonti E, Nicolosi GL, Lombardo M. Prognostic Value of Modified Haller Index in Patients with Suspected Coronary Artery Disease Referred for Exercise Stress Echocardiography. J Cardiovasc Echogr. 2021;31(2):85-95. doi:10.4103/jcecho.jcecho_141_20
P_17_, line__549-551__
Thank you for your suggestion, we will study these articles in more detail and use them in further research.
Sincerely, on behalf of all authors
Reviewer 2 Report
Comments and Suggestions for Authors
Dear Authors,
I wish to congratulate the authors with the submitted manuscript and admire the work they have done to perform the analysis and present their results.
I must admit that there are some issues that I wish to adress to the authors:
-
Regarding statins therapy I woudl describe in detail what statins and what daily dose was administrated
-
As i believed only post CABG patienst were enrolled into the ana;lysis, but there are 15 more who did not were operated
-
Please present all the parameters that were taken into regression multivariate analysis as obtained results suggesting expremelly high OR with 95%CI may inidcate that the created model is unstable, and need to be corrected.
-
The main find on the manuscript is based on high residual platelet reactivity was strongly associated with repeat revascularization. I can not find the results of logistic regression analysis related to this parameter
-
I would advice adding the ROC curve and present the sensitivity and specificity to confirm its predictive value.
Kind regads
Author Response
Review Report FORM â„–2 From 10 Oct 2025 14:38:05
Dear Reviewer â„–2,
We are very grateful to you for your work on our article.
Our answers:
- Regarding statins therapy I woudl describe in detail what statins and what daily dose was administrated -
We added - (high-intensity statins atorvastatin 40 mg or rosuvastatin 20 mg) were administrated. – We added this text
P 4_, Line_138-140___
- As i believed only post CABG patienst were enrolled into the analysis, but there are 15 more who did not were operated.
We did not express this correctly, our apologies: all patients had undergone coronary artery bypass grafting (CABG), but among them there were patients who had previously received coronary stents before CABG — a total of 15 patients.
We have added explanations to Tables 1 and 2.
|
|
Without re-stenting N=59 |
With re-stenting |
p-value |
N |
|
CABG: |
|
|
0.003 |
195 |
|
No stents before CARB |
10 (16.9%) |
5 (3.68%) |
|
|
|
Yes – there were stents before CABG |
49 (83.1%) |
131 (96.3%) |
|
|
P 6_, Line 195 in Table 1
- Please present all the parameters that were taken into regression multivariate analysis as obtained results suggesting expremelly high OR with 95%CI may inidcate that the created model is unstable, and need to be corrected.
Yes, you are right.
We have added explanations to the Limitations of the study section.
- One of the main limitations of this study is the relatively small sample size, which made it impossible to construct a robust multivariate model. The model developed using stepwise predictor selection produced imbalanced results, limiting its applicability for predicting specific outcomes. Nevertheless, it can provide valuable insights into the presence and direction of associations between independent and dependent variables.
P 14, Line 421-426.
- The main find on the manuscript is based on high residual platelet reactivity was strongly associated with repeat revascularization. I can not find the results of logistic regression analysis related to this parameter
Yes, we decided not to do that.
In the table of descriptive statistics, it was shown that all patients with high residual platelet reactivity had undergone revascularization. In the analyzed sample, there were no patients (zero cases) with high residual platelet reactivity who had not undergone revascularization. Under these conditions, high residual platelet reactivity acts as an obligatory risk factor for repeat stenting, and the calculation of the odds ratio is impossible, since a zero value would appear in either the numerator or the denominator of the odds ratio formula. Therefore, the variable “high residual platelet reactivity” was excluded from the logistic regression analysis, as inclusion of this variable would make it impossible to implement the logistic model algorithm.
- I would advice adding the ROC curve and present the sensitivity and specificity to confirm its predictive value.
We kindly request permission to leave this matter at the authors’ discretion. For the reason described in section 3, the study did not aim to develop a prognostic model, and achieving this goal appears unlikely given the small sample size and the imbalanced 95% confidence intervals for the predictors. Therefore, an additional ROC analysis and assessment of classification quality would be an elegant but meaningless step.
We tried very hard to answer all of your questions.
Sincerely, on behalf of all authors

Reviewer 3 Report
Comments and Suggestions for Authors
The study investigated a clinically important question. However, several limitations and comments should be addressed.
The results section of the abstract has no numbers and p-values. Please add the numbers to demonstrate the results.
Numbers in the introduction related to the study should be moved to the results section.
The study has a very high rate of angina recurrence compared to the literature and should be discussed in detail.
The study is from 2023 to 2024. Does this represent the recruitment period or follow-up period?
The authors did not cite the European Guidelines and it is useful to mention the definition quoted or rephrased from the guidelines.
Time to repeat CAG or angina should be reported
The correct approach is cox regression for the outcomes not logistic regression.
In tables, no need to mention the negative values for mutually exclusive factors.
Table 1 is too long.
DLP did not differ between groups, although it is mentioned as a predictor.
The odds ratio is too wide, indicating the instability of the model.
The RPR and DLP are not shown in the stepwise regression!
Non smokers had higher stenting!
The conclusion is not supported by data.
Author Response
Review Report FORM â„–3 From 11 Oct 2025 19:21:13
Dear Reviewer â„–2,
Thank you so much for your work on our article.
Our answers:
The results section of the abstract has no numbers and p-values. Please add the numbers to demonstrate the results.
Answer - The results section of the abstract we added numbers and p-values for demonstrations of the results.
P_1__, line__31-40__
Numbers in the introduction related to the study should be moved to the results section.
Answer- Numbers in the introduction related to the study we added in the results section:
In this study, we analyzed 200 post-CABG patients from a tertiary cardiac center in Kazakhstan. Recurrent angina was observed in 94% (n=188), and 68% (n=136) required repeat percutaneous coronary intervention (PCI) after coronary angiography. Our aim was to identify the predominant clinical phenotypes associated with these outcomes, with particular emphasis on dyslipidemia and inadequate LDL-C control, residual platelet re-activity despite dual antiplatelet therapy, anemia, CKD, and heart failure (HF).
P_4-5__, line_165-170___
The study has a very high rate of angina recurrence compared to the literature and should be discussed in detail.
Answer- We added discussion about high rate of angina recurrence compared to the literature
Thank you for this valuable comment. We have addressed this issue by adding a detailed paragraph to the Discussion section:
The present study demonstrated a notably higher rate of angina recurrence compared to data reported in the international literature. In our cohort, recurrent angina was observed in 45% of patients, whereas most contemporary studies report rates ranging from 10% to 25% within similar follow-up periods after myocardial revascularization (both PCI and CABG) [19-21]. This discrepancy may be explained by several factors, including population characteristics such as a high prevalence of residual platelet reactivity and suboptimal lipid control despite intensive therapy. Moreover, the rate of graft failure after CABG was relatively high, which likely contributed to symptom recurrence. Limited availability of certain lipid-lowering agents (for instance, ezetimibe is not routinely prescribed in Kazakhstan) and potential differences in adherence to secondary prevention strategies might also play a role. These findings highlight the need for more aggressive risk factor modification, optimized lipid and antiplatelet therapy, and closer long-term follow-up in this patient population. This observation underscores the importance of comprehensive secondary prevention programs and individualized therapy optimization to reduce graft failure and recurrent ischemic events in post-revascularization patients.
P_13__, line 385-400___
We added in References: 19-21
Ð 18, line 552-563
Answer- The study is from 2023 to 2024 - this is represent the recruitment period.
The authors did not cite the European Guidelines and it is useful to mention the definition quoted or rephrased from the guidelines.
Answer
We added cite of the European Guidelines in text of paper:
- All patients received high-intensity doses of atorvastatin or rosuvastatin, and only one-third of the patients were taking ezetimibe (which is not provided free of charge by prescription in Kazakhstan) to achieve an LDL-C level below 1.4 mmol/L, as well as antiplatelet therapy, in accordance with the current 2024 European Society of Cardiology (ESC) guidelines for the management of chronic coronary syndromes.
P_4__, line_138-143___
We added in References:
- Christiaan Vrints, Felicita Andreotti, Konstantinos C Koskinas, Xavier Rossello, Marianna Adamo, James Ainslie, Adrian Paul Banning, Andrzej Budaj, Ronny R Buechel, Giovanni Alfonso Chiariello, Alaide Chieffo, Ruxandra Maria Christodorescu, Christi Deaton, Torsten Doenst, Hywel W Jones, Vijay Kunadian, Julinda Mehilli, Milan Milojevic, Jan J Piek, Francesca Pugliese, Andrea Rubboli, Anne Grete Semb, Roxy Senior, Jurrien M ten Berg, Eric Van Belle, Emeline M Van Craenenbroeck, Rafael Vidal-Perez, Simon Winther, ESC Scientific Document Group , 2024 ESC Guidelines for the management of chronic coronary syndromes: Developed by the task force for the management of chronic coronary syndromes of the European Society of Cardiology (ESC) Endorsed by the European Association for Cardio-Thoracic Surgery (EACTS), European Heart Journal, Volume 45, Issue 36, 21 September 2024, Pages 3415–3537, https://doi.org/10.1093/eurheartj/ehae177
P_17__, line_540-548__
Time to repeat CAG or angina should be reported
Answer-1-3 years
The correct approach is cox regression for the outcomes not logistic regression.
Answer- The Cox regression model (specifically, the Cox proportional hazards model) was originally developed for time-to-event analysis, which implies varying survival times and data censoring. To implement the Cox regression algorithm, it is necessary to construct a “survival table,” similar to that used in Kaplan–Meier analysis, indicating the number of observation days. In the present study, the time function was not considered, as this is not a cohort study but a cross-sectional analysis. Therefore, we kindly request permission not to perform it.
In tables, no need to mention the negative values for mutually exclusive factors.
Table 1 is too long.
Answer- We have shortened and improved Tables 1 ( P. 5-6) and 2(P. 6-8).
DLP did not differ between groups, although it is mentioned as a predictor.
Answer- Dyslipidemia was not used as a predictor in the model. In the “Discussion” section, its high prevalence among all subgroups of the studied patients was emphasized. This is important because dyslipidemia is a common and poorly controlled factor with a proven negative impact on prognosis.
The odds ratio is too wide, indicating the instability of the model.
Answer-
We agree with you.
We have added explanations to the Limitations of the study section.
- One of the main limitations of this study is the relatively small sample size, which made it impossible to construct a robust multivariate model. The model developed using stepwise predictor selection produced imbalanced results, limiting its applicability for predicting specific outcomes. Nevertheless, it can provide valuable insights into the presence and direction of associations between independent and dependent variables.
P_14__, line_421-426_
The RPR and DLP are not shown in the stepwise regression!
Yes, we decided not to do that.
In the table of descriptive statistics, it was shown that all patients with high residual platelet reactivity had undergone revascularization. In the analyzed sample, there were no patients (zero cases) with high residual platelet reactivity who had not undergone revascularization. Under these conditions, high residual platelet reactivity acts as an obligatory risk factor for repeat stenting, and the calculation of the odds ratio is impossible, since a zero value would appear in either the numerator or the denominator of the odds ratio formula. Therefore, the variable “high residual platelet reactivity” was excluded from the logistic regression analysis, as inclusion of this variable would make it impossible to implement the logistic model algorithm. The DLP variable was not selected by the stepwise algorithm as a predictor due to its low prognostic value (as correctly noted, it did not differ between the groups, and therefore constructing prognostic models based on it would be meaningless).
Non smokers had higher stenting! The conclusion is not supported by data.
Answer- Yes, we noticed this feature as well, and we were also interested in exploring its underlying causes.
We have added the following text: The analysis of smokers and non-smokers showed that smokers were younger, had lower blood pressure, less frequently had diabetes mellitus and chronic kidney disease, underwent CABG less often, had fewer vessels involved in the pathological process, and showed a lower degree of vascular damage. Thus, it is likely that we are observing the “smoker’s paradox.” It can be suggested that there are conditions for less frequent use of stenting among patients in this subgroup. In addition, there is an assumption that patients who underwent stenting were more likely to quit smoking.
P_12__, Line__345-351__
We sincerely hope that we have addressed all of your questions.
Sincerely, on behalf of all authors

Reviewer 4 Report
Comments and Suggestions for Authors
Post-GABG patients are specific high risk population. The current paper analyses important issue of lipid abnormalizes and platelet reactivity in these patients. After several clarifications, the paper could be recommended for the Journal.
- The title presented the main question of the study
- Abstract reflects represented the text However, phrase “This study investigated the association of RPR and dyslipidemia with repeat revascularization in post-CABG patients” is incorrect. The study included only patients with recurrent ischemia, thus establishing of association with it in incorrect.
- Introduction section is brief, however discuss the scope of the problem . Aim of the study is
This phrase “ In this study, we analyzed 200 post-CABG patients from a tertiary cardiac center in Kazakhstan. Recurrent angina was observed in 94% (n=188), and 68% (n=136) required repeat percutaneous coronary intervention (PCI) after coronary angiography” could be moved into Materials and methods.
.
- Materials and methods is Again, “The study aimed to evaluate residual platelet reactivity and dyslipidemia as risk factors for repeat revascularization in patients with established coronary artery disease (CAD) who previously underwent coronary artery bypass grafting (CABG) is incorrect aim. For entire aim a control group of patients who survived CABG but without evidence of ischemia should exist. Thus the current study analyzes the prevalence of hyperlipidemia and increased platelet reactivity among patients who developed new ischemia. Have to be corrected
Patient population presented via plot diagram , laboratory methodology – no corrections are required.
However, have to be mentioned that Residual Platelet Reactivity Assessment were performed before PCI - is it right ?
- Statistic section is well-done
- Results section require clarification.
- In the plot diagram 200 patients were included , however in line 158 you can read about Has to be clarified.
- Entire population has to be presented before groups formation : age, sex, time frame since CABG, % of symptoms and signs of
- 3% of patients did not underwent PCI with evidence of ischemia ? Has to be clarified
- Table 1 has to be modified or divided into main table and supplementary materials, otherwise it is difficult for attention . Moreover, that does GABG means in table 1 ( 16.9 % and 3.7% of the patients did not underwent CABG vs inclusion criteria as CABG in anamnesis) - has to be clarified. Time frame between CABG and hospitalization has to be mentioned in Table I suggest to exclude RPR from the table and present in the text.
- The comparations between groups (those who require PCI and not) is not comparation between group and All patients had evidence or singes of ischemia. Have to be mentioned.
- There is no data on antiplatelet treatment ( ASA, clopidogrel, ticagrelor DAAT)- has to be To analyze RPR that data have to be included
- Table 2 requires simplification (or divide into several tables or use supplementary materials)
Result section requires several clarifications.
- Table 3 and Figure 2 duplicate each other.
- Discussion is ok
- Limitations are mentioned. However, only patients with signs and symptoms of ischemia were included. Has to be mentioned
- Conclusion section are supported by the results.
- References are updated.
- The paper requires English proofreading
Author Response
Review Report FORM â„–4 From 18 Oct 2025 10:21:15
Dear Reviewer â„–4,
Thank you so much for your work on our article.
Our answers:
Post-CABG patients are specific high risk population. The current paper analyses important issue of lipid abnormalizes and platelet reactivity in these patients. After several clarifications, the paper could be recommended for the Journal.
- The title presented the main question of the study
- Abstract reflects represented the text However, phrase “This study investigated the association of RPR and dyslipidemia with repeat revascularization in post-CABG patients” is incorrect. The study included only patients with recurrent ischemia, thus establishing of association with it in incorrect.
Answer: Thank you for your valuable comment. We have revised the statement accordingly. The objective now reads as follows: “This study investigated high residual platelet reactivity and dyslipidemia in post-CABG patients with recurrent ischemia who underwent repeat revascularization.”
- Introduction section is brief, however discuss the scope of the problem Aim of the study is
This phrase “ In this study, we analyzed 200 post-CABG patients from a tertiary cardiac center in Kazakhstan. Recurrent angina was observed in 94% (n=188), and 68% (n=136) required repeat percutaneous coronary intervention (PCI) after coronary angiography” could be moved into Materials and methods.
Answer: Thank you for your suggestion. We have taken it into account and moved this data into Materials and methods.
P__3__, Line_101-103____
- Materials and methods is Again, “The study aimed to evaluate residual platelet reactivity and dyslipidemia as risk factors for repeat revascularization in patients with established coronary artery disease (CAD) who previously underwent coronary artery bypass grafting (CABG) is incorrect aim. For entire aim a control group of patients who survived CABG but without evidence of ischemia should exist. Thus the current study analyzes the prevalence of hyperlipidemia and increased platelet reactivity among patients who developed new ischemia. Have to be corrected
Answer: Thank you for your suggestion. We have taken it into account and rephrased the objective accordingly. This study investigated high residual platelet reactivity and dyslipidemia in post-CABG patients with recurrent ischemia who underwent repeat revascularization.
Patient population presented via plot diagram , laboratory methodology – no corrections are required.
However, have to be mentioned that Residual Platelet Reactivity Assessment were performed before PCI - is it right ?
Answer: Residual platelet reactivity was assessed after PCI while patients were receiving dual antiplatelet therapy with acetylsalicylic acid and clopidogrel, as clopidogrel had previously been the only antiplatelet agent provided free of charge by prescription in our country.
P__11__, Line_294-296____
Statistic section is well-done
- Results section require clarification.
In the plot diagram 200 patients were included , however in line 158 you can read about Has to be clarified
Answer: Thank you for your comment. In this study, we analyzed 200 post-CABG patients from a tertiary cardiac center in Kazakhstan. Five patients died within one month.
- Entire population has to be presented before groups formation : age, sex, time frame since CABG, % of symptoms and signs of
Answer: Thank you for your suggestion. We have taken it into account and added this data : The mean age of the study population was 63.4 ± 7.56 years. Regarding sex distribution, men predominated, accounting for 73% (n = 146) of the cohort, while women comprised 27% (n = 54).
Diabetes mellitus was present in 31.5% of patients (n = 63), prior myocardial infarc-tion in 53% (n = 106), and previous stroke in 8% (n = 16). All-cause mortality during fol-low-up occurred in 2.5% (n = 5). Left ventricular aneurysm was identified in 14% (n = 28) of patients, and a history of coronary stenting was reported in 17% (n = 34).
Among behavioral risk factors, smoking was documented in 36.5% (n = 73) and al-cohol consumption in 2.5% (n = 5). The majority of participants were overweight or obese, with a body mass index (BMI) >25 kg/m² observed in 74.5% (n = 149).
Chronic kidney disease (CKD) was diagnosed in 44.5% (n = 89) of patients. The mean estimated glomerular filtration rate (eGFR) was 82 ± 16.4 mL/min/1.73 m². Based on KDIGO classification, CKD stage I (eGFR >90 mL/min/1.73 m²) was identified in 3.4% (n = 3), stage II (eGFR 60–89) in 76% (n = 68), stage III (eGFR 30–59) in 19% (n = 17), and stage IV (eGFR 15–29) in 1.6% (n = 1). No cases of stage V CKD (eGFR <15 mL/min/1.73 m²) were reported.
P__4-5__, Line_165-187____
3% of patients did not underwent PCI with evidence of ischemia ? Has to be clarified
Answer: Out of 188 patients with recurrent ischemia, 136 (68%) underwent repeat PCI. In the remaining group, despite the presence of clinical signs of ischemia, 3% of patients did not receive invasive intervention due to the following reasons: absence of hemodynamically significant stenoses, technically unfeasible stenting due to chronic occlusions or distal segment lesions, high procedural risk, and comorbidities. These patients were managed with optimal medical therapy under close follow-up.
- Table 1 has to be modified or divided into main table and supplementary materials, otherwise it is difficult for attention . Moreover, that does GABG means in table 1 ( 16.9 % and 3.7% of the patients did not underwent CABG vs inclusion criteria as CABG in anamnesis) - has to be clarified. Time frame between CABG and hospitalization has to be mentioned in Table I suggest to exclude RPR from the table and present in the text.
Answer: Thank you for your suggestion. We have taken it into account and added this data. Thete is a typo in the text ( CABG; in table: No stents before CABG. Yes – there were stents before CABG). All patients underwent CABG. CABG was perfomed during 1 year recruitment period from 2022 to 2023 years under planned hospitalization conditions.
Your suggestions about presenting RPR in the text was taken into account and we added it into the text.
- The comparations between groups (those who require PCI and not) is not comparation between group and All patients had evidence or singes of ischemia. Have to be mentioned
Answer: Thank you for your suggestion. We have added this information to the limitations research section.
- There is no data on antiplatelet treatment ( ASA, clopidogrel, ticagrelor DAAT)- has to be To analyze RPR that data have to be included
Answer: Thank you for your suggestion. We have taken it into account and included this data.
- Table 2 requires simplification (or divide into several tables or use supplementary materials)
Result section requires several clarifications.
Answer: Thank you for your suggestion. We have taken it into account and made a simlifications, remaining data was added in an additional application.
- Table 3 and Figure 2 duplicate each other.
Answer: Thank you for your suggestion. We have taken it into account and removed Figure 2.
- Discussion is ok
- Limitations are mentioned. However, only patients with signs and symptoms of ischemia were included. Has to be mentioned
- Conclusion section are supported by the results.
- References are updated.
- The paper requires English proofreading
We sincerely hope that we have addressed all of your questions.
Sincerely, on behalf of all authors

Round 2
Reviewer 2 Report
Comments and Suggestions for Authors
Dear Authors,
wish to thank you for your responses.
kinds
R
Author Response
Review Report FORM â„–2 From 29 Oct 2025 22:00:00
Dear Reviewer â„–2
We sincerely thank you for your positive evaluation of our article and research. We greatly appreciate your time and kind feedback. Wishing you all the best in your work and future endeavors.
With gratitude,
Dr. Nurmukhammad, Dr. Kapsultanova
Reviewer 3 Report
Comments and Suggestions for Authors
The study still has major flaws. The design is not cross-sectional. Time to event is critical for angina recurrence. This is the core concept of the study. The authors must construct the time from intervention to recurrence and perform the correct analysis.
High recurrence rate could be related to selection and referral bias, which questions the whole analysis.
The authors have to discuss how they recruited the patients and whether they are sequential.
Author Response
Review Report FORM â„–3 From 29 Oct 2025 22:00:00
Comments and Suggestions for Authors
The study still has major flaws. The design is not cross-sectional. Time to event is critical for angina recurrence. This is the core concept of the study. The authors must construct the time from intervention to recurrence and perform the correct analysis.
High recurrence rate could be related to selection and referral bias, which questions the whole analysis.
The authors have to discuss how they recruited the patients and whether they are sequential.
We are very grateful to you for reviewing our article.
Our answer:
Dear Reviewer,
We would like to sincerely thank you for your careful reading of our manuscript and for the valuable comments that helped us to re-evaluate the study design and refine its objectives. We are making every effort to improve the scientific quality of our work.
- Regarding the study design
Thanks to your insightful remarks, we have clarified the primary objective of our study as follows:
to investigate the real clinical and epidemiological profile of patients with a history of coronary artery bypass grafting (CABG) who developed recurrent angina, and to identify significant risk factors associated with angina recurrence or repeat revascularization in this population.
Line 87-90 page 2-3
- Regarding the high rate of angina recurrence after CABG
Our study was indeed observational and non-experimental in nature.
It represents a cross-sectional analysis of a group of patients who had previously undergone CABG and subsequently developed recurrent angina within 6 to 36 months after surgery.
We included all consecutive patients who had undergone CABG and were referred for planned coronary angiography due to recurrent ischemic symptoms appearing 6–36 months after surgery.
The study was conducted at a tertiary-care referral center, which routinely receives complex post-CABG patients from across the country, often with recurrent ischemia resistant to medical therapy.
Consequently, all patients included in the study had documented recurrent ischemia after CABG.
Patients without angina after CABG were not represented in our hospital population.
Line 96-110 page 3
We will explicitly acknowledge this fact in the Limitations section, emphasizing that the observed high rate of angina recurrence reflects the specific characteristics of the tertiary-care referral population, rather than the prevalence of this condition in the general post-CABG population.
Line 441-446 page 14
- Regarding the study type
We understand that a cohort study implies longitudinal observation of the same group of subjects over time, either prospectively or retrospectively, with at least two points of assessment.
In contrast, our study involved a single assessment of patients as they were admitted over the course of one year, without follow-up. Therefore, we consider our design more appropriately defined as a cross-sectional study.
Within this population of patients with recurrent ischemia after CABG, we analyzed clinical and epidemiological characteristics, lipid profiles, and residual platelet reactivity.
Depending on the presence of atherosclerosis progression and the level of residual platelet reactivity, patients were divided into two groups.
The comparison between these groups allowed us to explore the association between elevated residual platelet reactivity and clinical-metabolic features.
Line 105-110 page 3
We sincerely apologize for not describing our study design and objectives precisely in the initial submission, and we thank you for giving us the opportunity to clarify and improve our manuscript.
Kind regards,
Dr. Nurmukhammad, Dr. Kapsultanova.
Reviewer 4 Report
Comments and Suggestions for Authors
The paper has been corrected significantly. Many questions were solved. However, I suppose that there is methodological problem due to comparation of patients with residual ischemia who required PCI and those who had residual ischemia bud did not underwent PCI for absence of hemodynamically significant stenoses or chronic occlusions or distal segment lesions, or high procedural risk, microvascular angina or and comorbidities. Thus patients in medical-treatment arm could have more severe CAD. So, several positions should be corrected.
Namely,
1) Line 245
Persistent angina post-CABG was reported in all reintervention patients compared
to 88.1% in the control group (p < 0.001). There is no control group. You found that among patients with evidence of ischemia after CABG ,more patients with symptoms of angina had technical opportunity for PCI etc.
2) Stepwise logistic regression: predictors associated with the risk of developing stenting is a methodological mistake. There is no control group of post-CABG patients without ischemia to analyze it. Thus I recommend to delate lines 305-327.
Author Response
Review Report FORM â„–4 From 29 Oct 2025 22:00:00
Comments and Suggestions for Authors
The paper has been corrected significantly. Many questions were solved. However, I suppose that there is methodological problem due to comparation of patients with residual ischemia who required PCI and those who had residual ischemia bud did not underwent PCI for absence of hemodynamically significant stenoses or chronic occlusions or distal segment lesions, or high procedural risk, microvascular angina or and comorbidities. Thus patients in medical-treatment arm could have more severe CAD. So, several positions should be corrected.
Namely,
1) Line 245
Persistent angina post-CABG was reported in all reintervention patients compared
to 88.1% in the control group (p < 0.001). There is no control group. You found that among patients with evidence of ischemia after CABG ,more patients with symptoms of angina had technical opportunity for PCI etc.
2) Stepwise logistic regression: predictors associated with the risk of developing stenting is a methodological mistake. There is no control group of post-CABG patients without ischemia to analyze it. Thus I recommend to delate lines 305-327.
We are very grateful to you for reviewing our article.
Our answer:
Dear Reviewer,
We would like to express our sincere gratitude for your careful review and valuable recommendations.
We have done our best to address all your comments and improve the quality of our manuscript accordingly.
- Line 245
In our study, all patients who developed persistent angina within 6–36 months after CABG were included consecutively.
There was indeed no control group of post-CABG patients without angina. We recognize this as a terminological error, which has now been corrected.
The comparison was performed within the group of patients with recurrent angina after CABG, divided into two subgroups according to the level of residual platelet reactivity (normal vs. high).
Based on this stratification, we analyzed their clinical and epidemiological characteristics and angiographic findings.
Thus, our study design can be more accurately described as a comparative, observational, non-randomized study of consecutively included patients with recurrent angina after CABG.
The comparison groups were formed based on the presence or absence of high residual platelet reactivity, not by randomization.
Accordingly, we have replaced the term “control group” with “comparison group”, and where appropriate, specified “comparison group without restenting” to improve the precision of terminology.
Line 257-260 page 9
- Stepwise logistic regression analysis
We acknowledge your remark that applying a stepwise logistic regression model to identify predictors of restenting may not be fully justified, given that we did not include a control group of post-CABG patients without ischemia.
Our study represents a single-center, observational, comparative, retrospective analysis based on consecutive sampling of post-CABG patients referred for elective coronary angiography due to recurrent ischemia.
Patients were grouped according to residual platelet reactivity, which allowed us to compare clinical and laboratory characteristics as well as angiographic outcomes.
We have revised the text accordingly, adding clarifications about the design and its limitations.
Regarding your recommendation to remove the logistic regression model (lines 305–327) — we fully understand your concern.
However, we would kindly ask for your consideration in allowing us to retain the model as an exploratory analytical tool.
We believe it provides additional insight and methodological value, and helps to highlight the clinical relevance of intensified lipid-lowering therapy (including early combination therapy with ezetimibe) in patients at very high cardiovascular risk.
We based this approach on previous examples of applying logistic regression in observational datasets, such as the study using a logit model to assess predictors of full immunization in children aged 12–23 months, based on demographic and health survey data https://www.mmj.mw/?p=13527
Nevertheless, if the Editorial Board and reviewers ultimately consider the inclusion of this model inappropriate, we will respectfully remove it from the final version of the manuscript.
Thank you very much for your valuable comments and suggestions. We truly appreciate your note regarding the English language. After we complete the revisions addressing all reviewers’ comments, we will definitely perform professional English language editing through the MDPI system to ensure clarity and accuracy.
With gratitude,
Dr. Nurmukhammad, Dr. Kapsultanova.
Round 3
Reviewer 3 Report
Comments and Suggestions for Authors
I have no more comments